# Treatment of *Plasmodium falciparum* merozoites with the protease inhibitor E64 and mechanical filtration increases their susceptibility to complement activation

José A. Stoute[1,2]*, Mary E. Landmesser[1], Sergei Biryukov[2¤]

1 The Division of Infectious Diseases, Department of Medicine, the Penn State College of Medicine, Hershey, Pennsylvania, United States of America, 2 Department of Microbiology and Immunology, The Penn State College of Medicine, Hershey, Pennsylvania, United States of America

¤ Current address: USAMRIID, Fort Detrick, Maryland, United States of America
* jstoute@psu.edu, jastoute@gmail.com

**Data Availability Statement:** All relevant data are within the paper and its Supporting Information files.

## Abstract

*Plasmodium falciparum* malaria killed 451,000 people in 2017. Merozoites, the stage of the parasite that invades RBCs, are a logical target for vaccine development. Treatment with the protease inhibitor E64 followed by filtration through a 1.2 μm filter is being used to purify merozoites for immunologic assays. However, there have been no studies to determine the effect of these treatments on the susceptibility of merozoites to complement or antibodies. To address this gap, we purified merozoites with or without E64 followed by filtration through either a 1.2 or 2.7 μm filter, or no filtration. Merozoites were then incubated in either 10% fresh or heat-inactivated serum followed by surface staining and flow cytometry with monoclonal antibodies against the complement effector molecules C3b or C5b9. To determine the effect of anti-merozoite antibodies, we incubated merozoites with MAb5.2, a mouse monoclonal antibody that targets the merozoite surface protein 1. We used an amine-reactive fluorescent dye to measure membrane integrity. Treatment with E64 resulted in an insignificant increase in the proportion of merozoites that were C3b positive but in a significant increase in the proportion that were C5b9 positive. Filtration increased the proportion of merozoites that were either C3b or C5b9-positive. The combination of filtration and E64 treatment resulted in marked deposition of C3b and C5b9. MAb5.2 induced greater complement deposition than serum alone or an IgG2b isotype control. The combination of E64 treatment, filtration, and MAb5.2 resulted in very rapid and significant deposition of C5b9. Filtration through the 1.2 μm filter selected a population of merozoites with greater membrane integrity, but their integrity deteriorated rapidly upon exposure to serum. We conclude that E64 treatment and filtration increase the susceptibility of merozoites to complement and antibody. Filtered or E64-treated merozoites are not suitable for immunologic studies that address the efficacy of antibodies *in vitro*.

**Funding:** JAS T2017-274 Global Health Information Technology Fund www.ghitfund.org The funders had no role in study design, data collection and analysis, decision to publish, or preparation of the manuscript.

**Competing interests:** The authors have declared that no competing interests exist.

## Introduction

*Plasmodium falciparum* was responsible for 451,000 deaths in 2017. Although a vaccine was recently licensed in Europe, its efficacy is still less than optimal. Thus, much more work is needed to develop an effective malaria vaccine. One major target of vaccine development is the merozoite stage of the parasite that invades red blood cells (RBCs). Investigations of the interaction of merozoites with RBCs and the identification of the protective immune responses of the host are keys to this endeavor. The traditional approach to studying the interaction between merozoites and RBCs has been to purify late stage infected RBCs (IRBCs) from culture and allow them to develop naturally, eventually leading to the egression of merozoites and invasion of surrounding uninfected RBCs. This approach has the limitation of not allowing the isolation of the different stages of invasion since invasion occurs asynchronously and shortly after egression [1]. To circumvent this limitation, Boyle et al. [2] developed a technique to obtain a high yield of pure merozoites by synchronizing late stage parasites using the protease inhibitor trans-Epoxysuccinyl-L-leucylamido(4-guanidino) butane (E64) to a stage prior to merozoite egression followed by filtration through a 1.2 μm filter. This technique, or a variation of it, is now being used to understand the mechanisms of RBC invasion [3, 4], study anti-merozoite immunity [5–12], and study the susceptibility of parasites to anti-malarial drugs [13]. In addition, this technique is being applied to the study of other parasites such as *P. yoelii* [14]. However, despite its rapid adoption to the study of merozoite biology and immunity, there has been no study of the effects of treatment with E64 and of mechanical filtration on the susceptibility of merozoites to immune-mediated effector mechanisms. Thus, in order to fill this gap in our knowledge, we decided to study the effect of E64 treatment and filtration on the viability and complement susceptibility of merozoites.

## Materials and methods

### Parasite strains, cultures, and serum

*P. falciparum* strain 3D7 was obtained from BEI Resources (BEI Resources, Manassas, VA). Parasites were maintained in heat-inactivated (HI) 0.5% AlbuMax I (ThermoFisher Scientific, Waltham, MA) in RPMI 1640 (Sigma-Aldrich, St. Louis, MO) containing 200 μg/ml hypoxanthine, and 10 μg/ml gentamicin with 2% hematocrit O+ RBCs in an atmosphere of 5% $O_2$, 5% $CO_2$, and 90% $N_2$ [15]. The cultures were synchronized at least weekly with 5% sorbitol [16]. Serum was collected as described previously [17]. The human serum and RBCs were obtained from a single donor under a protocol approved by the Penn State College of Medicine Institutional Review Board. Heat inactivation was carried out by incubation at 56 ˚C for 30 minutes.

### Purification of merozoites, measurement of membrane integrity, and deposition of C3b and C5b9

The day prior to the assay, late stage parasites were purified by Percoll gradient centrifugation [18]. From a 50 ml culture, a typical recovery was 1–3 x $10^8$ late stage IRBCs at close to 100% purity. After washing with plain RPMI 1640, IRBCs were resuspended in 8–12 ml of 0.5% HI AlbuMAX I (ThermoFisher) in RPMI 1640 without RBCs. One half of the culture was incubated with 10 μM E64 (Sigma-Aldrich). After overnight (~12 hrs) incubation using standard conditions, the culture containing E64 was resuspended in fresh media without E64 and incubated for one additional hour at 37 ˚C. Following this incubation, each culture was divided into three equal portions. One portion was used without further processing (unfiltered), a second portion was filtered through a 1.2 μm filter (Pall Corporation, Ann Arbor, MI), and the third through a 2.7 μm filter (GE Healthcare, Waukesha, WI). The cultures were then

distributed equally into wells of a 96-well plate. The plate was centrifuged at 1,100 xg for 15 min and the pellets were resuspended in 100 μL of either plain RPMI (baseline samples), RPMI containing 10% heat-inactivated human serum (HIS), or in RPMI 1640 containing 10% fresh human serum (FS). Immediately after resuspension, EDTA was added to the baseline samples to a final concentration of 10 mM. The remaining wells were incubated at 37 ˚C for 0, 15, or 30 min in 10% HIS or FS containing media. The time "0" wells were resuspended in media followed immediately by addition of EDTA to a final concentration of 10 mM. The same was done for the 15- and 30-minute time points at the conclusion of each incubation period. Following the addition of EDTA, the samples were apportioned equally into three wells of a separate 96-well plate. After another round of centrifugation at 1,100 xg, the pellets were resuspended in PBS containing 1:1000 dilution of far red live-dead reagent (Thermo-Fisher) containing a 1:5000 dilution of Hoechst 33342 (ThermoFisher), with or without the addition of FITC-labeled monoclonal antibody 7C12 (a kind gift from Dr. Ronald Taylor) at 10 μg/ml that targets human C3b/iC3b [19] or FITC-labeled aE11 (Hycult Biotech, Wayne, PA) at 0.1 μg/ml that targets a neoantigen of C9 in C5b9, also known as the membrane attack complex (MAC) [20]. The live-dead reagent measures membrane integrity by binding to amines on the surface of cells with intact membranes and to amines in both the cytosol and surface of cells with compromised membranes. Thus, cells with compromised membranes will appear brighter than cells with intact membranes. After an incubation of 1 hour at room temperature, the plates were centrifuged and the pellets were resuspended in 50 μl 2% paraformaldehyde and stored at 4˚C until acquisition.

## Effect of anti-merozoite monoclonal antibody MAb5.2 on merozoite integrity and complement susceptibility

Mouse monoclonal antibody IgG$_{2b}$ MAb5.2 was raised against the 19 kDa subunit of the merozoite surface protein 1 (MSP1$_{19}$) [21] and was purified from hybridoma supernatant (ATCC, Manassas, VA) to study the effect of anti-merozoite antibodies. A nonspecific IgG2b mouse monoclonal antibody was used as negative control (Santa Cruz Biotechnology, Dallas, TX). Both antibodies were used at 50 μg/ml in the presence of HIS or FS as described above.

## Flow cytometry and gating strategy

To compare the density of merozoites across groups we calculated the merozoite event rate based on the number of events during the period of acquisition. To compare this number across samples we always used the same dilution by adding 25 μL of sample to 100 μL of PBS. The samples were acquired in an LSRII (Becton Dickinson, NJ). S1 Fig summarizes our gating strategy. Samples were acquired using logarithmic amplification (S1A Fig). Single merozoites were gated based on their low forward and side scatter profile and Hoechst positive signal (S1B Fig). Staining with Alexa 488-conjugated MAb5.2 confirmed that the merozoites in this gate were MSP1 positive (S1C Fig). After an additional incubation in the live-dead reagent, the population in the merozoite gate was then examined in a plot of FITC vs APC (S1D Fig). A histogram of the APC positive events revealed a bright and a dim population corresponding to membrane-damaged and membrane intact merozoites respectively (S1E Fig). When the FITC-labeled anti-C3b or anti-C5b9 was added, the percent C3b or C5b9-positive merozoites could be easily determined (S1F Fig).

## Statistical analysis

All experiments underwent at least three independent repetitions. Statistical analysis was done using Sigmaplot v14.0 (Systat Software Inc., San José, CA). To compare the means of two

groups we used paired t-test or the Wilcoxon Signed Rank Test for non-parametric data. To compare the means of two groups to a control we used the repeated measures one-way analysis of variance with Holm-Sidak test for multiple comparisons. All tests were two-tailed with $\alpha <$ 0.05.

## Results

### Effect of E64 and filtration on C3b and C5b9 deposition in the absence of anti-merozoite antibodies

Deposition of C3b and C5b9 was mostly observed on membrane-damaged merozoites in the presence of FS. Deposition was minimal in the presence of HIS (see Effect of Antibody MAb5.2 below). Because deposition of C3b and C5b9 seemed to occur very rapidly upon exposure to FS, we incubated merozoites in plain RPMI in order to determine the baseline deposition prior to addition of serum. Use of E64 resulted in a small increase in the proportion of C3b-positive merozoites, which was statistically significant only for unfiltered merozoites at 30 min (Fig 1A and 1B). However, the use of filtration, both in the presence or absence of E64 resulted in statistically significant increases in C3b-positive merozoites (Fig 1C and 1D). On the other hand, treatment with E64 resulted in statistically significant increases in C5b9-positive merozoites at 15 and 30 min in both unfiltered and filtered merozoites (Fig 2A and 2B). Similarly to C3b, filtration resulted in an increased proportion of C5b9-positive merozoites at all time points (Fig 2C and 2D). In order to assess the combined effect of filtration and E64-treatment on complement deposition, we compared the C3b and C5b9 deposition on untreated unfiltered merozoites to E64-treated filtered merozoites. Fig 3 shows that the combination of filtration and E64 treatment resulted in marked increases in the proportion of C3b and C5b9-positive merozoites.

### Effect of E64 and filtration on merozoite cell membrane integrity in the absence of antibodies

We found that filtration through the 1.2 μm filter consistently resulted in a lower proportion of merozoites with membrane damage at baseline regardless of E64 treatment status (S2 Fig). However, the integrity of these merozoites deteriorated rapidly both in FS and HIS to the point that at time 0 there was no significant difference in the proportion of membrane-damaged merozoites between filtered and unfiltered merozoites. Treatment with E64 resulted in consistent trends towards increased damaged merozoites in HIS (S3A and S3B Fig). In the presence of FS there was also a trend towards more membrane-damaged merozoites that was only statistically significant for unfiltered merozoites at time points 15 and 30 min (S3C Fig) and for merozoites filtered through the 1.2 μm filter at 15 min (S3D Fig). When we compared the integrity of unfiltered merozoites to that of filtered E64-treated merozoites in FS we again saw that merozoites filtered through the 1.2 μm had a lower proportion of merozoites with membrane damage at baseline (Fig 4). However, once again, the difference disappeared by time 0 and after 30 min of incubation a greater proportion of filtered merozoites showed membrane damage than unfiltered merozoites (Fig 4).

### Effect of anti-merozoite monoclonal antibody MAb5.2 on complement deposition

Since both filtration and E64 treatment during merozoite purification enhance complement deposition on merozoites, we wanted to ascertain if these merozoites were more susceptible to classical complement pathway activation. To this end, we used the anti-merozoite antibody

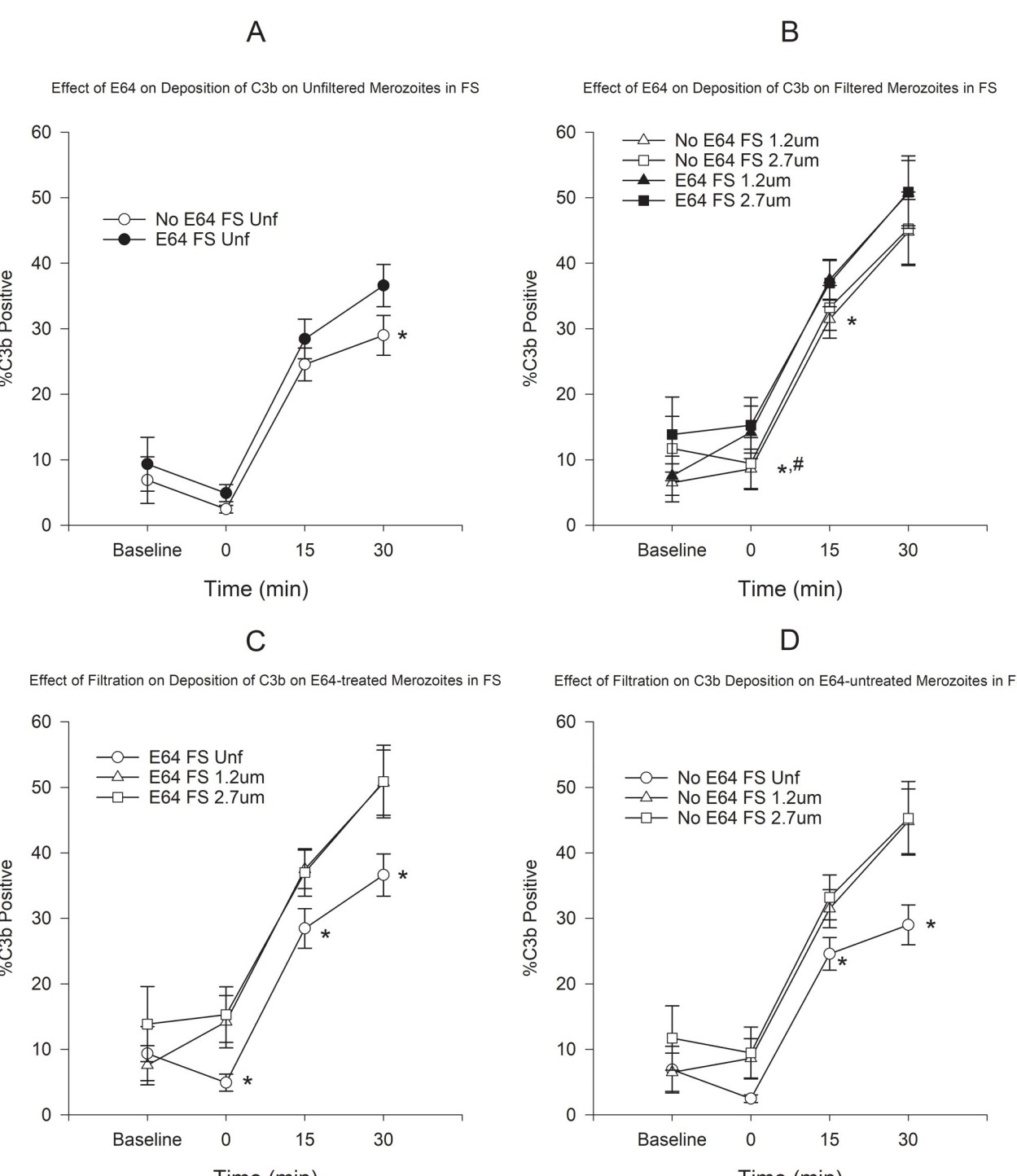

**Fig 1. Effect of filtration and E64 treatment on the deposition of C3b on merozoites in FS.** A) Effect of E64 treatment on deposition of C3b on unfiltered merozoites. B) Effect of E64 treatment on deposition of C3b on filtered merozoites. C) Effect of filtration on deposition of C3b on E64-treated merozoites. D) Effect of filtration on deposition of C3b on E64-untreated merozoites. *$P < 0.05$ for the comparison of unfiltered to filtered merozoites (Panel A, C, and D). *$P < 0.05$ for the comparison between 1.2 μm filtered merozoites and #$P < 0.05$ for the comparison between 2.7 μm filtered merozoites (Panel B). Unf = unfiltered, FS = fresh serum, and HIS = heat-inactivated serum. Error bars represent standard errors of the mean.

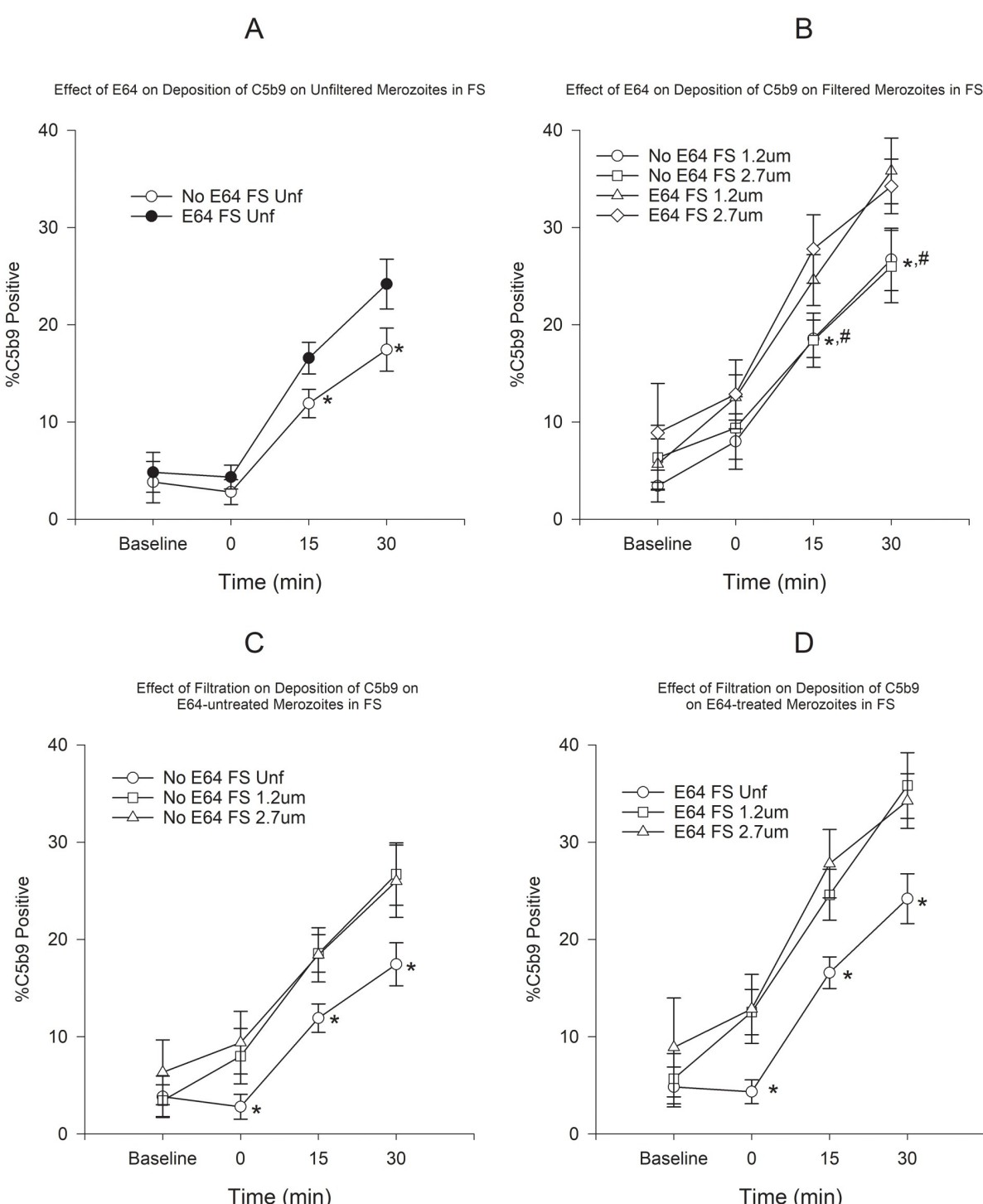

**Fig 2. Effect of E64 treatment and filtration on the deposition of C5b9 on merozoites in FS.** A) Effect of E64 on deposition of C5b9 on unfiltered merozoites. B) Effect of E64 on deposition of C5b9 on filtered merozoites. C) Effect of filtration on deposition of C5b9 on E64-treated merozoites. D) Effect of filtration on deposition of C5b9 on E64-untreated merozoites. $^*P < 0.05$ for the comparison of unfiltered to filtered merozoites (Panel A, C, and D). $^*$ P < 0.05 for the comparison between E64-treated and untreated 1.2 μm-filtered merozoites and $^#P < 0.05$ for the comparison between E64-treated and untreated 2.7 μm-filtered merozoites (Panel B). Error bars represent standard errors of the mean.

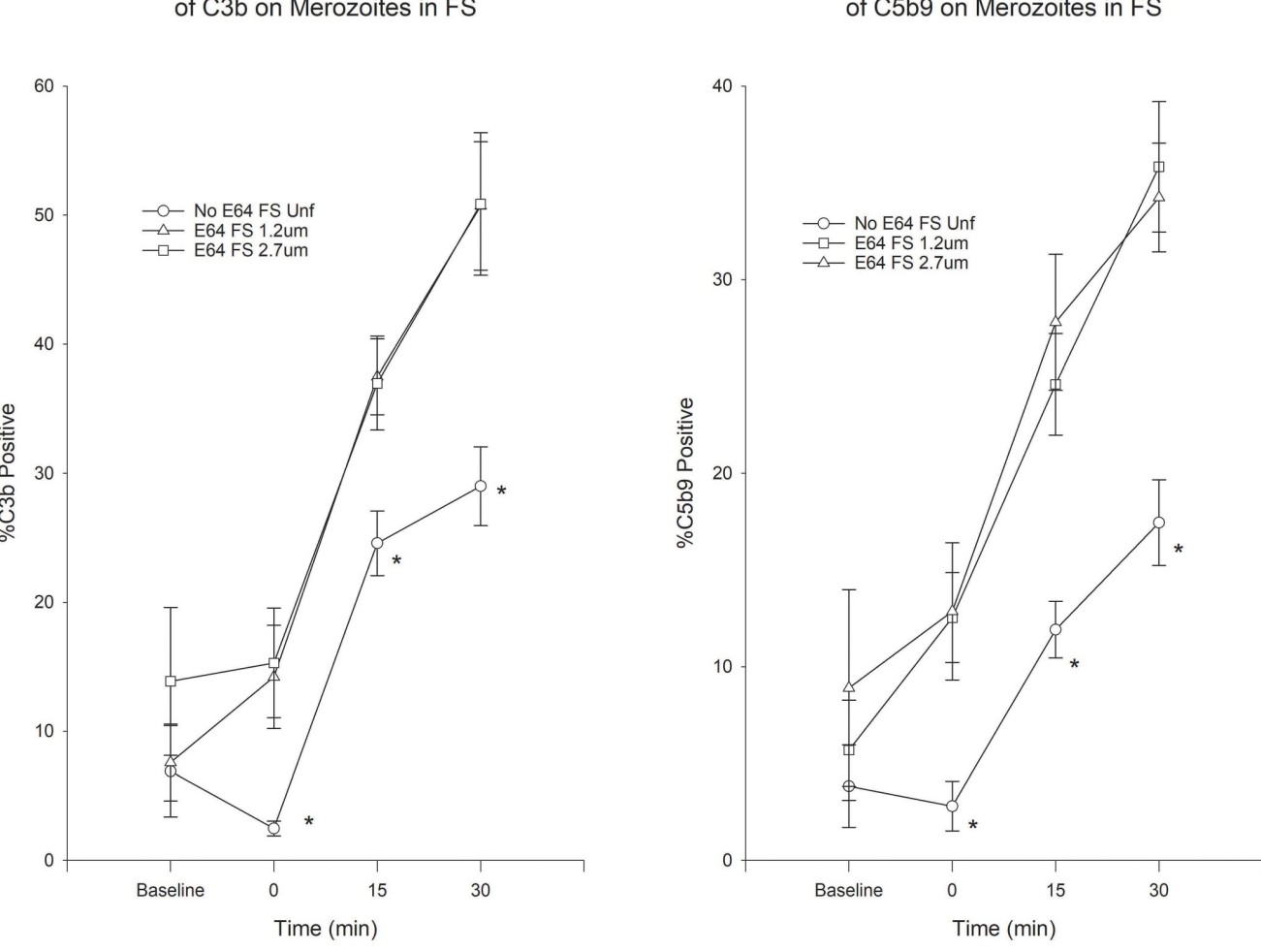

**Fig 3. Effect of combined E64 treatment and filtration on the deposition of C3b and C5b9 on merozoites.** A) C3b deposition on merozoites. B) C5b9 deposition on merozoites. *P < 0.05 for the comparison of unfiltered to filtered merozoites. Error bars represent standard errors of the mean.

MAb5.2 to simulate the effect of anti-merozoite antibodies on the membrane integrity and complement deposition of merozoites. Incubation with MAb5.2 in FS resulted in a significant increase in the proportion of C3b-positive merozoites that far exceeded what was seen with the IgG2b control antibody both in the presence or absence of E64 (Fig 5). As expected, incubation in HIS produced no significant change in the proportion of C3b-positive merozoites. Filtration and E64 treatment resulted in small increases in the proportion of C3b-positive merozoites but the increases were not statistically significant (Figs 5 and 6). The results of C5b9 deposition paralleled the results of C3b (Figs 7 and 8). When we compared unfiltered E64-untreated to filtered E64-treated merozoites, we saw that the latter showed a greater proportion of C3b-positive merozoites at all time points but a statistically significant difference was found only at time 30 min (Fig 9A). On the other hand, filtration in combination with E64 treatment resulted in a significantly marked increase in C5b9-positive merozoites at all time points (Fig 9B). Filtration and E64 treatment of merozoites resulted in increased classical complement pathway activation, with C3b and C5b9 deposition, in the presence of anti-merozoite antibody and FS.

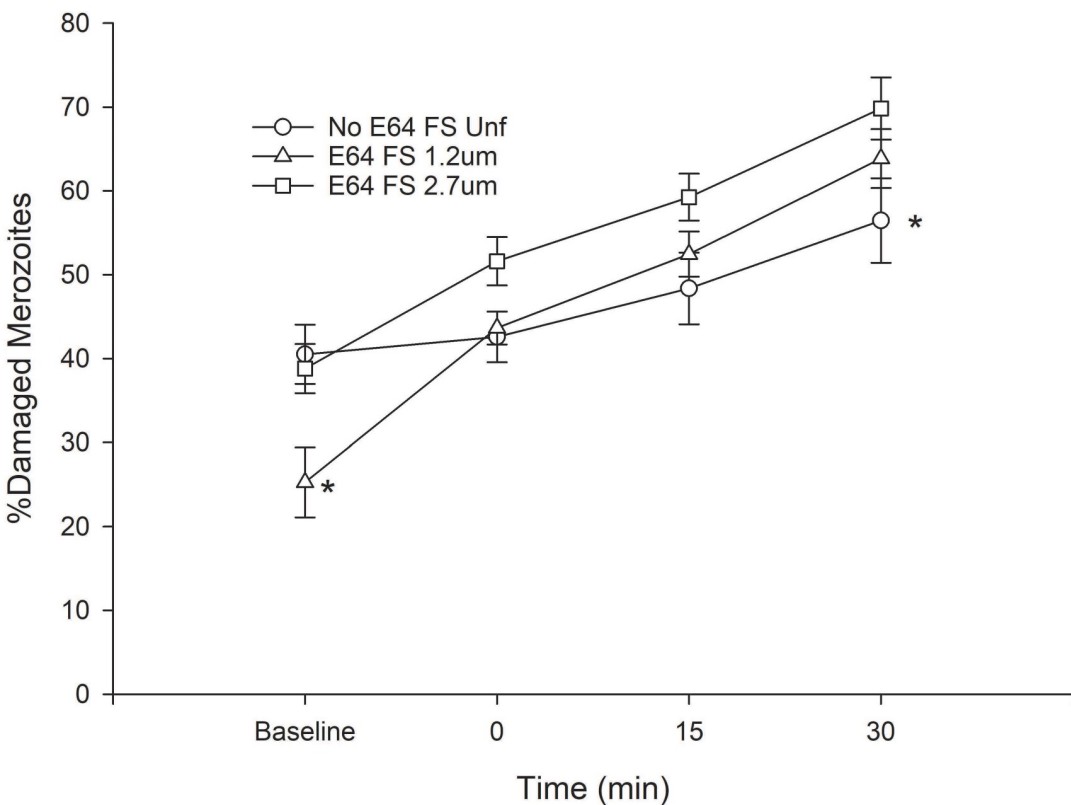

**Fig 4. Effect of combined E64 treatment and filtration on merozoite membrane integrity in FS.** *P < 0.05 for the comparison between 1.2 μm filtered and unfiltered merozoites at baseline and unfiltered and filtered merozoites at time 30 min. Error bars represent standard errors of the mean.

### Effect of MAb5.2 on the membrane integrity of merozoites

We also examined membrane integrity of merozoites that were filtered and/or E64-treated to further evaluate the effect of anti-merozoite antibody in the presence of complement using live-dead stating. Incubation with MAb5.2 resulted in greater proportion of membrane-damaged merozoites both in the presence of FS and HIS, and regardless of filtration or E64 treatment, than with the control IgG2b antibody (S4 Fig). Filtration resulted in a trend towards increased proportion of membrane-damaged merozoites in the absence of E64 (S4A and S4B Fig) but this trend disappeared in the presence of E64 (S4C and S4D Fig). Treatment with E64 resulted in a trend towards a greater proportion of membrane-damaged merozoites (S5A and S5B Fig) but this trend was less obvious in filtered merozoites (S5C and S5D Fig). Comparison of unfiltered merozoites to E64-treated filtered merozoites incubated in the presence of MAb5.2 showed that the latter had a greater proportion of membrane-damaged merozoites at all time points, but the differences were not statistically significant (Fig 10).

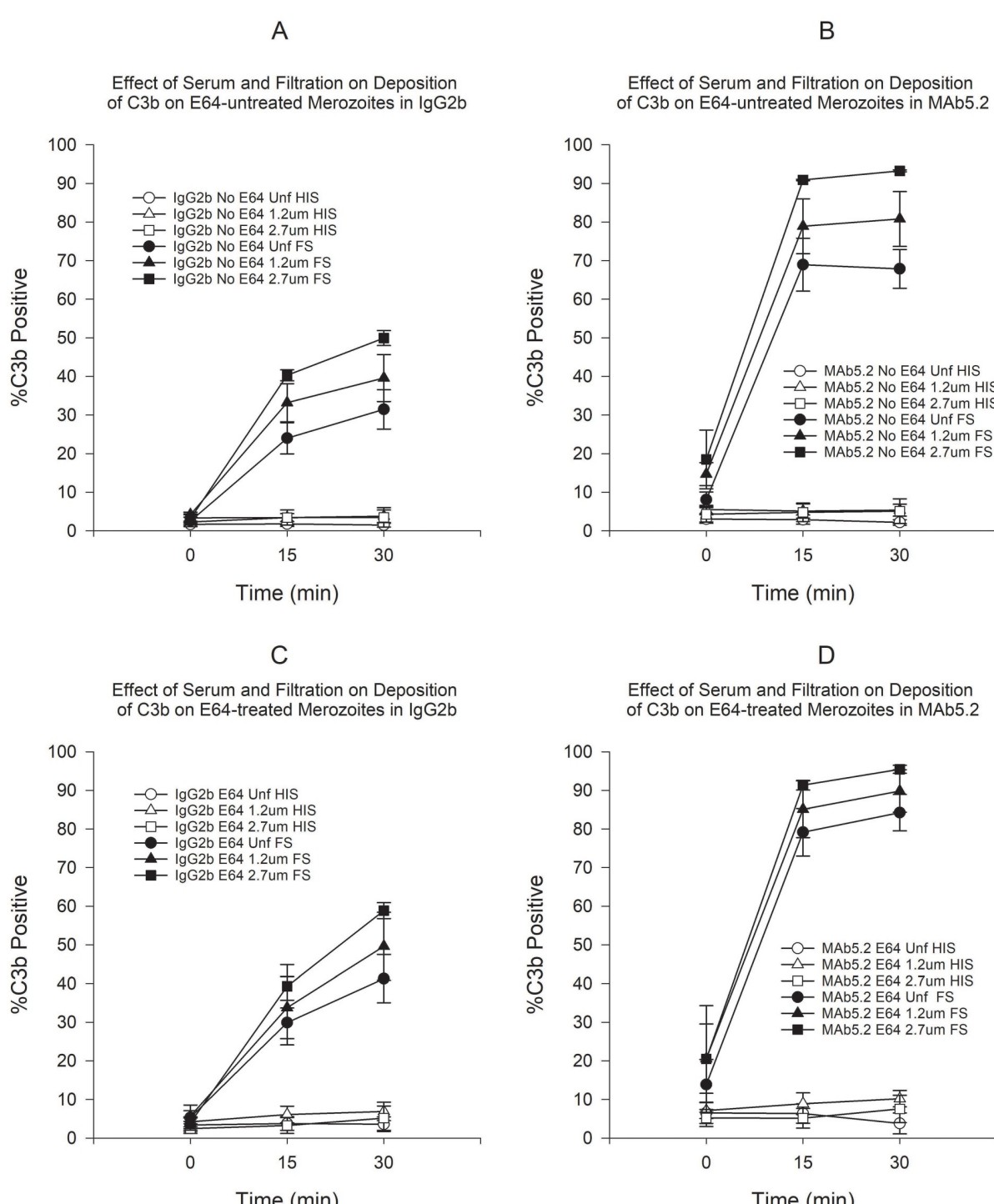

**Fig 5. Effect of serum and filtration on deposition of C3b on merozoites in either MAb5.2 or IgG2b.** A) Effect of serum and filtration on deposition of C3b on E64-untreated merozoites in IgG2b. B) Effect of serum and filtration on deposition of C3b on E64-untreated merozoites in MAb5.2. C) Effect of serum and filtration on deposition of C3b on E64-treated merozoites in IgG2b. D) Effect of serum and filtration on deposition of C3b on E64-treated merozoites in MAb5.2. Error bars represent standard errors of the mean.

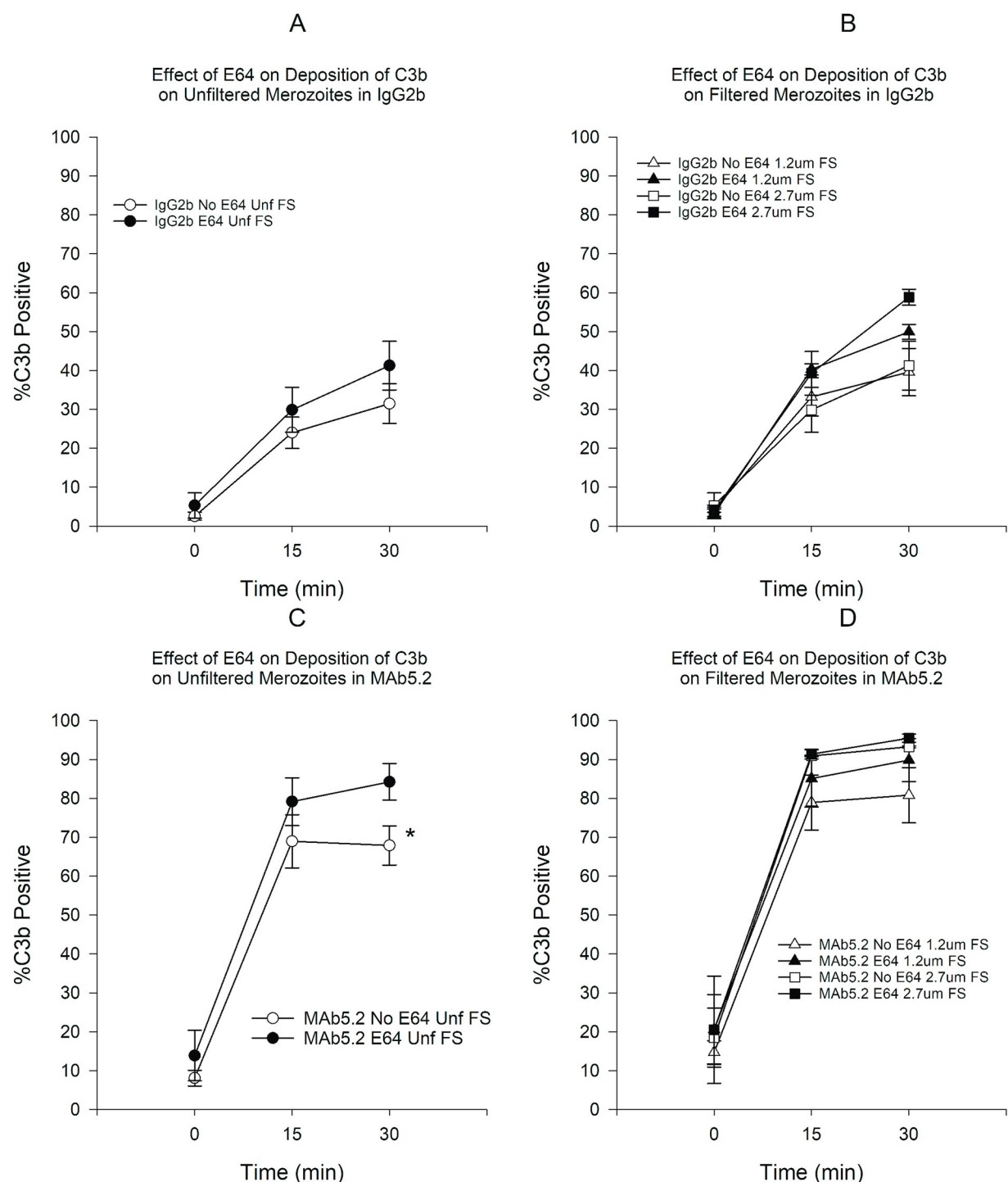

**Fig 6. Effect of E64 treatment on the deposition of C3b on merozoites in the presence of either MAb5.2 or IgG2b in FS.** A) Effect of E64 on deposition of C3b on unfiltered merozoites in the presence of IgG2b. B) Effect of E64 on deposition of C3b on filtered merozoites in the presence of IgG2b. C) Effect of E64 on deposition of C3b on unfiltered merozoites in the presence of MAb5.2. D) Effect of E64 on deposition of C3b on filtered merozoites in the presence of MAb5.2. $^*P < 0.05$. Error bars represent standard errors of the mean.

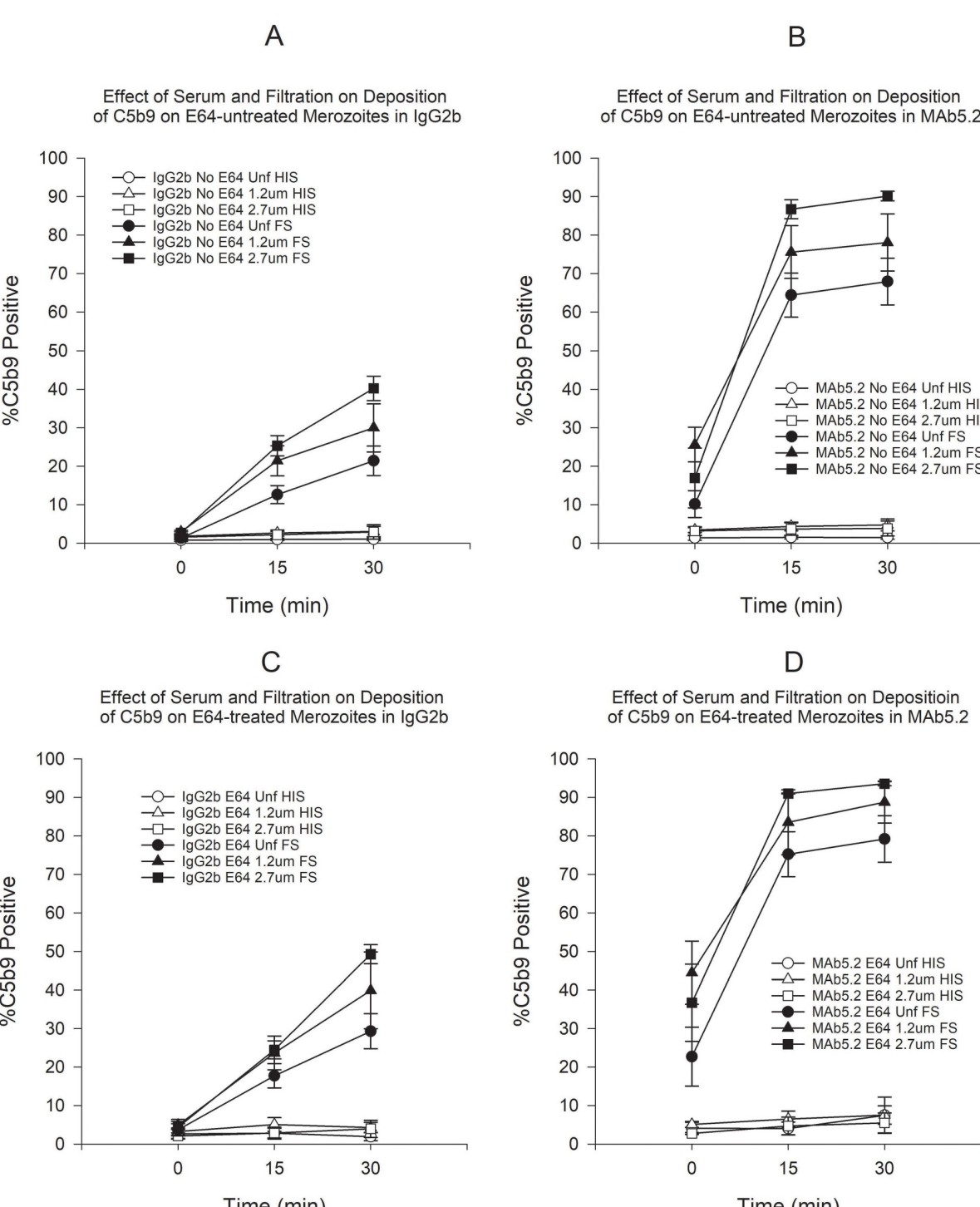

**Fig 7. Effect of serum and filtration on deposition of C5b9 on merozoites in either MAb5.2 or IgG2b.** A) Effect of serum and filtration on deposition of C5b9 on E64-untreated merozoites in the presence of IgG2b. B) Effect of serum and filtration on deposition of C5b9 on E64-untreated merozoites in the presence of MAb5.2. C) Effect of serum and filtration on deposition of C5b9 on E64-treated merozoites in the presence of IgG2b. D) Effect of serum and filtration on deposition of C5b9 on E64-treated merozoites in the presence of MAb5.2. Error bars represent standard errors of the mean.

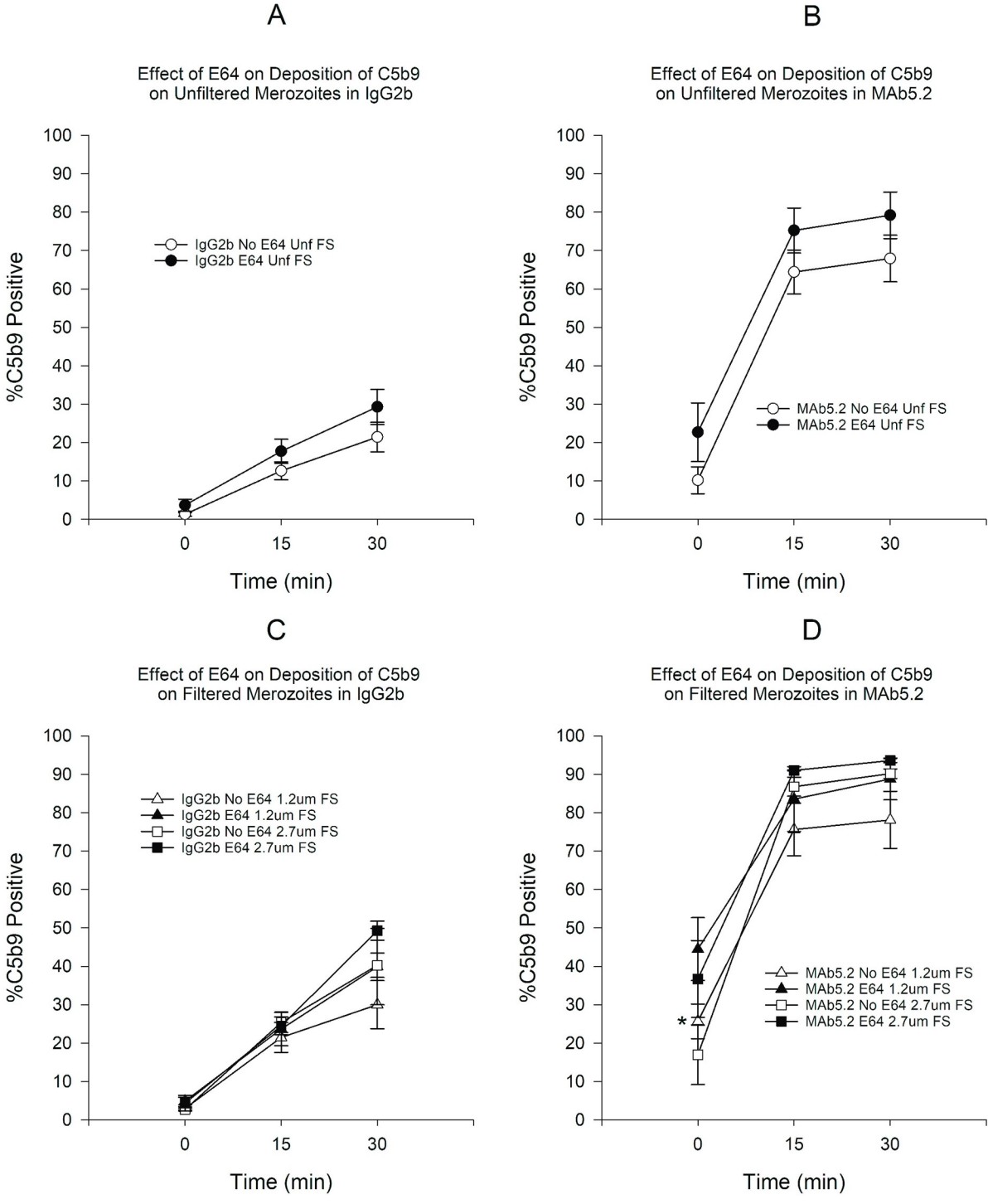

**Fig 8. Effect of E64 treatment on deposition of C5b9 on merozoites in the presence of either MAb5.2 or IgG2b in FS.** A) Effect of E64 on deposition of C5b9 on unfiltered merozoites in the presence of IgG2b. B) Effect of E64 on deposition of C5b9 on deposition of C5b9 on unfiltered merozoites in the presence of MAb5.2. C) Effect of E64 on deposition of C5b9 on filtered merozoites in the presence of IgG2b. D) Effect of E64 on deposition of C5b9 on filtered merozoites in MAb5.2. *P < 0.05 for the comparison between 1.2 μm filtered merozoites. Error bars represent standard errors of the mean.

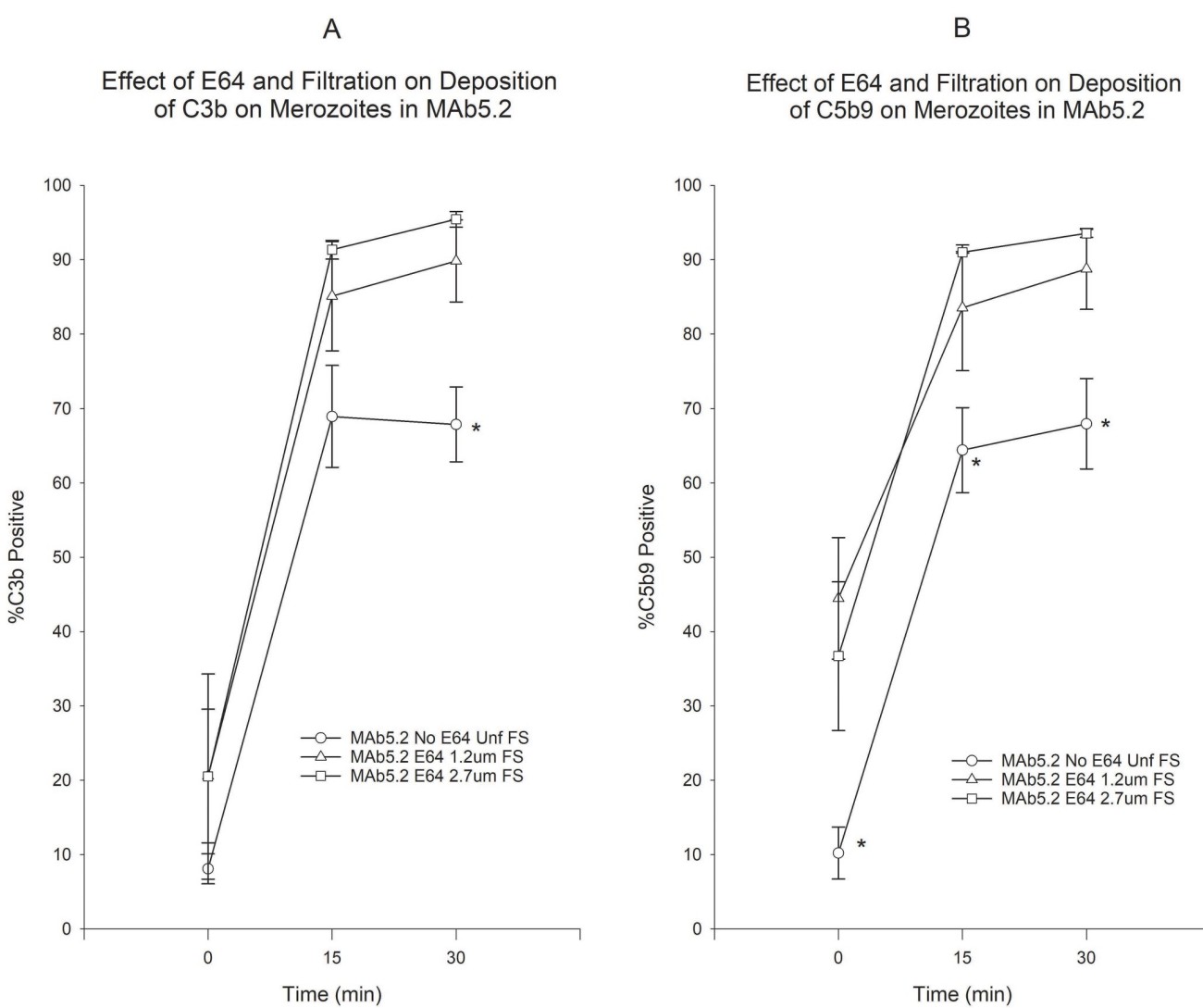

**Fig 9. Effect of combined E64 and filtration on deposition of C3b and C5b9 on merozoites in the presence of FS.** A) Effect of E64 and filtration on deposition of C3b on merozoites in the presence of MAb5.2. B) Effect of E64 and filtration on deposition of C5b9 on merozoites in the presence of MAb5.2. Error bars represent standard errors of the mean.

### Differential analysis based on Hoechst staining

Hoechst staining showed at least two populations of merozoites (S1 Fig), one with low staining and several with higher staining. We carried out a differential analysis of the two populations (S6 Fig). S7 Fig shows that high Hoechst merozoites have a higher proportion of membrane-damaged merozoites than low Hoechst merozoites. Therefore, differential Hoechst penetration is the most likely explanation for the difference in staining. In both populations we observed that filtration with the 1.2um filter selects a population of merozoites with greater membrane integrity as we had seen before. S8 Fig shows that the two populations showed increased deposition of C3b and C5b9 with the combination of E64 treatment and filtration with the 1.2um filter compared to E64-untreated unfiltered merozoites.

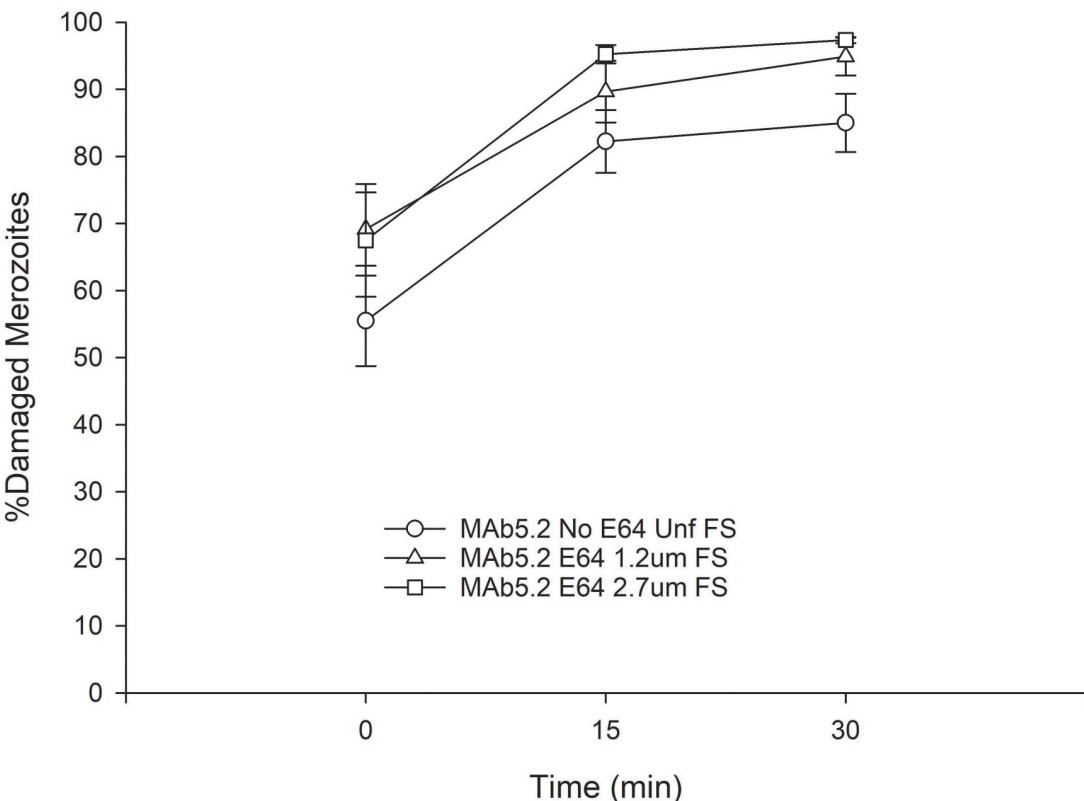

**Fig 10. Effect of E64 and filtration on the membrane integrity of merozoites in the presence of MAb5.2 and FS.** Error bars represent standard errors of the mean.

### Effect of filtration and E64 treatment on event rate of merozoites

We used the merozoite event rate as a proxy for merozoite density since the event rate is a function of the density. To do this we insured that the samples were divided equally among all wells and all the dilutions were the same prior to flow cytometry. S9 Fig shows that filtration with the 1.2 μm filter resulted in marked reduction in the merozoite event rate. This was further reduced by the use of E64. Incubation in FS or HIS had little or no effect on the event rate, suggesting that there was no significant lysis. Treatment with E64 led to clumping of merozoites which was obvious microscopically and by flow cytometry (S10 Fig), resulting in increased side scatter signal of merozoites.

### Discussion

We investigated the effect of E64 and filtration on the susceptibility of *P. falciparum* merozoites to complement deposition. Our results suggest that the use of E64 in combination with filtration results in increased complement susceptibility of merozoites. The effect seems to be mainly due to filtration although the use of E64 did result in increased in C5b9 deposition independently of filtration.

Although we tried to follow the technique originally reported by Boyle et. al. [2], we introduced some important differences. The principal difference between our method and that of Boyle et. al. [2] is that we chose to use Percoll gradient centrifugation to purify late trophozoites and schizonts as opposed to magnetic columns. Percoll purification is less expensive than magnetic purification and, in our hands, has always resulted in highly pure and viable late trophozoites and schizonts. It is unlikely that this change could be responsible for our findings since Percoll purification was used in all groups and the increases in complement activation were clearly seen in those groups where filtration and/or E64 were used. Another difference is that our parasite cultures were maintained in HI Albumax I as opposed to 10% plasma. We chose to use HI Albumax I because in preliminary experiments we observed high background complement deposition on merozoites grown in HI plasma or in Albumax I. The use of HI Albumax I resulted in minimal complement deposition at baseline and allowed us to compare the effect of the treatments without high background complement deposition. We incubated our schizonts with E64 for ~12hrs which is longer than the 8 hrs Boyle et al. [2] reported. It is unclear whether this difference is important and no studies have been done to determine the effects of the duration of incubation on merozoite viability. Hill et al. [6] recommended an incubation range of 6–10 hrs. Finally, in addition to using a 1.2 µm filter, we used an alternative filter with a 2.7 µm pore size. The rationale for this was to determine if increased pore size could avoid some of the deleterious effects of the smaller pore size.

We used MAb5.2, which is a mouse $IgG_{2b}$ monoclonal antibody raised against $MSP1_{19}$, to simulate the effect of human anti-merozoite antibodies. MAb5.2 was selected because it is plentiful in our laboratory and easily activates human complement [17, 22]. Incubation with MAb5.2 resulted in a rapid and large increase in the deposition of C3b and C5b9. We observed that the major effect of combining E64 treatment and filtration with MAb5.2 was a marked and rapid increase in C5b9-positive merozoites.

Since C5b9 or MAC acts by forming a pore in the cell membrane, we measured the membrane integrity of merozoites. Paradoxically, we observed that filtration through the 1.2 µm filter selected for merozoites with greater membrane integrity. The effect was also seen to some extent in merozoites filtered through the 2.7 µm filter but less predictably. The ability of filtration to select merozoites with intact membranes suggests that merozoites with damaged membranes swell to a larger size than merozoites with intact membranes. However, despite their greater membrane integrity, the membrane integrity of merozoites filtered through the 1.2 µm filter deteriorated rapidly upon exposure to serum to the point that at time 0 there was no difference between filtered and unfiltered merozoites. By contrast, the proportion of unfiltered damaged merozoites changed little between baseline and time 0. Thus, although a greater proportion of merozoites filtered through the 1.2 µm filter had intact membranes, these merozoites seemed to be more fragile and more sensitive to complement. The deterioration in membrane integrity of merozoites filtered through the 1.2 µm at time 0 is most likely due to complement deposition, as demonstrated by the significant increase in C3b and C5b9 deposition on these merozoites at time 0. From time 0 onward we did not see any significant differences in membrane integrity between filtered and unfiltered merozoites even though we did see very significant increase in the proportion of C3b and C5b9-positive filtered merozoites at all time points. One possible explanation is that the live-dead reagent may not have the dynamic range to detect these differences. In other words, after a certain minimum number of pores are formed the dye penetrates completely. E64 treatment resulted in a trend towards a higher proportion of merozoites with membrane damage, but this was statistically significant only in unfiltered merozoites incubated in FS at time 15 and 30 min.

MAb5.2 induced a rapid deterioration in membrane integrity in the presence of FS. Although there was a trend towards increased proportion of membrane-damaged merozoites

with filtration and E64 treatment, these differences were not statistically significant. These results again suggest that the live-dead reagent may not have the dynamic range to show differences in membrane integrity at this level of MAC deposition.

We attempted to investigate whether the differences we saw in terms of complement deposition between unfiltered merozoites and filtered E64-treated merozoites impacted their ability to invade RBCs. However, our attempts were unsuccessful in that none of the merozoite preparations, regardless of treatment status, were able to invade. Instead, what we observed was extensive attachment to RBCs (S11 Fig). Boyle et al. [2] reported that while most of the merozoites egressed after removal of E64, they were unable to invade. Filtration selected a population with high invasion capacity. This is consistent with our observation that filtered merozoites had better membrane integrity than unfiltered merozoites at baseline. They also reported that invasion decreased dramatically within 10 min of purification by incubation at 37 °C but it was improved by keeping the merozoites at room temperature until incubation which is the approach we followed. However, even at room temperature the invasion capacity was less than 20% after one hour. Both filtration and E64 treatment resulted in marked decrease in the numbers of free merozoites (S9 Fig) which makes invasion even more difficult to detect. It is possible that pre-incubation in E64-free medium may have decreased the invasion capacity of the merozoites. Yet, even when we removed this step, invasion was negligible (S11 Fig). It is difficult to make comparisons to the Boyle et al. [2, 5] data since no absolute numbers are given in any of their studies. Their methods require large volumes of culture and it is possible that they overcame the low invasion capacity of filtered merozoites by increasing their numbers. Based on their report, an extremely high ratio of purified merozoites to RBCs (~6:1) is needed to attain a parasitemia of ~1.8% after 24/48 h (S5B Fig, [2]). Assuming that 16 merozoites egress from one schizont, this merozoite to RBC ratio is equivalent to incubating target RBCs with a starting parasitemia of ~27.3% (3 infected RBCs for 8 uninfected RBCs). By contrast, using intact schizonts at a parasitemia of 0.5 to 1% we can easily achieve a new invasion parasitemia of ~8%.

Several other groups have reported the use of filter-purified merozoites. The earliest report we could find was that of Mrema et al. [23] who purified merozoites by use of a 1.2 μm filter but did not assess their viability. Sterling [24] estimated the invasiveness of filtered *P. chabaudi* merozoites, a rodent malaria species, to between 1 and 5%. Recently, Mutungi et al. [14] studied the effect of pre-incubation temperature on the invasion capacity of filtered *P. yoelii* merozoites. They reported that pre-incubation at 37 °C following filtration resulted in rapid deterioration of invasive capacity within 30 min. Deterioration was slowed by pre-incubation at 23 °C and at 15 °C, and the latter resulted in the best preservation of invasive capacity. Kit et al. [13] reported a ring parasitemia of 0.07 to 0.15% with the use of filtered merozoites. Most recently, Kumar et al. [25] compared the invasiveness of γ-irradiated and non-irradiated merozoites filtered through a 2 μm polycarbonate filter. They reported that the invasion efficiency of unirradiated filtered merozoites was close to 0% [25]. Therefore, filtered merozoites have an intrinsic inability to invade RBCs although their invasiveness could be improved by pre-incubation at lower than room temperature.

The mechanism of increased complement deposition following filtration and E64 treatment is unclear. Recent studies have shown that merozoites are capable of recruiting complement regulators such as C1q esterase inhibitor (C1INH) and Factor H (FH) to their surface [26–28]. This recruitment appears to be the result of direct interaction of these complement regulators with the surface merozoite ligands MSP3.1 and Pf92 respectively [26–28]. It is tempting to speculate that filtration could strip the merozoites of some of these ligands leading to decreased recruitment of complement regulators. It is unlikely that the filters themselves are activating complement because the addition of serum takes place after filtration. Although E64 seemed

to have a lesser contribution to the increase in complement susceptibility, we did observe that in combination with filtration there was at least an additive effect, especially for C5b9. E64 is an irreversible cysteine protease inhibitor that inhibits the food vacuole protease falcipain and it is also an autophagy inhibitor [29, 30]. In addition, E64 inhibits the final step in the release of merozoites, the rupture of the RBC membrane, trapping merozoites [31]. It is possible that E64 may inhibit processing of merozoite surface proteins that are ligands for complement regulators or that are involved in complement regulation themselves.

## Conclusions

Our results clearly show that filtration in combination with E64 treatment increases the susceptibility of merozoites to complement-mediated damage both in the absence and presence of anti-merozoite antibodies. This finding has significant implications since a number of groups have used filter-purified merozoites to measure the inhibitory and opsonophagocytic activity of anti-merozoite antibodies [5–12]. Boyle et al. [5] reported that antibodies from individuals living in a malaria endemic region could prevent invasion of E64-treated filtered merozoites in a complement-dependent manner with a major contribution from C1q. We did not measure C1q, but, given our results, these findings should be interpreted with caution.

## Supporting information

**S1 Fig. Gating strategy.** A) Sample of merozoites filtered through a 1.2 um filter and incubated in FS for 15 min was acquired using logarithmic amplification on side scatter vs. forward scatter plot. B) Single merozoites were gated on the basis of low side scatter and positive Hoechst staining as well as on positive staining with Alexa 488-conjugated MAb5.2 (C). D) Dot plot of measurement of merozoite membrane integrity. E) Histogram of merozoites from panel D. F) After addition of FITC-labeled anti-C3b or anti-C5b9 antibodies, we measured the percent FITC positive population. EXP-18-FJ5475.
(DOCX)

**S2 Fig. Effect of filtration on merozoite membrane integrity.** A) Effect of filtration on membrane integrity of E64-treated merozoites in FS. B) Effect of filtration on membrane integrity of E64-untreated merozoites in FS. C) Effect of filtration on membrane integrity of E64-treated merozoites in HIS. D) Effect of filtration on membrane integrity of E64-untreated merozoites in HIS. *P < 0.05 for the comparison between unfiltered and 1.2 μm filtered merozoites (Panels A and B), and for the comparison between unfiltered and filtered merozoites (Panels C and D). Error bars represent standard errors of the mean.
(DOCX)

**S3 Fig. Effect of E64 treatment on merozoite membrane integrity.** A) Effect of E64 treatment on membrane integrity of unfiltered merozoites in HIS. B) Effect of E64 treatment on membrane integrity of filtered merozoites in HIS. C) Effect of E64 treatment on membrane integrity of unfiltered merozoites in FS. D) Effect of E64 treatment on membrane integrity of filtered merozoites in FS. *P < 0.05 for the comparison between E64-treated and untreated merozoites (Panel C) and the comparison between 1.2 μm filtered merozoites (Panel D). Error bars represent standard errors of the mean.
(DOCX)

**S4 Fig. Effect of serum and filtration on the membrane integrity of merozoites in IgG2b or MAb5.2.** A) Effect of serum and filtration on the membrane integrity of E64-untreated merozoites in the presence of IgG2b. B) Effect of serum and filtration on the membrane integrity of E64-untreated merozoites in the presence of MAb5.2. C) Effect of serum and filtration on the

membrane integrity of E64-treated merozoites in the presence of IgG2b. D) Effect of serum and filtration on the integrity of E64-treated merozoites in the presence of MAb5.2. Error bars represent standard errors of the mean.
(DOCX)

**S5 Fig. Effect of E64 and serum on the integrity of merozoites in the presence of IgG2b or MAb5.2.** A) Effect of E64 and serum on the membrane integrity of unfiltered merozoites in the presence of IgG2b. B) Effect of E64 and serum on the membrane integrity of unfiltered merozoites in the presence of MAb5.2. C) Effect of E64 and serum on the membrane integrity of filtered merozoites in the presence of IgG2b. D) Effect of E64 and serum on the membrane integrity of filtered merozoites in the presence of MAb5.2. Error bars represent standard errors of the mean.
(DOCX)

**S6 Fig. Effect of E64 and filtration on merozoite event rate.** A) Effect of filtration on event rate of E64-untreated merozoites in FS. B) Effect of filtration on event rate of E64-untreated merozoites in HIS. C) Effect of filtration on event rate of E64-treated merozoites in FS. D) Effect of filtration on event rate of E64-treated merozoites in HIS.
(DOCX)

**S7 Fig. E64 causes clumping of merozoites.** Following Percoll enrichment schizonts were incubated overnight in complete media containing HIS with or without 10 uM E64. After removal of E64, merozoites were allowed to egress. Panel B shows increased proportion of clumped merozoites compared to panel A, which was confirmed by microscopic examination. EXP-18-FJ5475.
(DOCX)

**S8 Fig. Measurement of invasion.** Free merozoites purified by filtration through a 1.2 um filtered were incubated in complete media containing 10% HIS and RBCs at 0.5% hematocrit for a few seconds (A) or 15 minutes (B). The RBCs were washed once and incubated in complete medium overnight. The following day a sample was obtained for flow cytometry and stained with Hoechst 33342. The apparent percent IRBCs increased from 2.5% to 3.6%. However, microscopic examination showed that the vast majority of these invasive forms were merozoites attached to the surface of RBCs. EXP-18-FJ5497.
(DOCX)

**S9 Fig. Effect of E64 and filtration on merozoite event rate.** A) Effect of filtration on event rate of E64-untreated merozoites in FS. B) Effect of filtration on event rate of E64-untreated merozoites in HIS. C) Effect of filtration on event rate of E64-treated merozoites in FS. D) Effect of filtration on event rate of E64-treated merozoites in HIS.
(DOCX)

**S10 Fig. E64 causes clumping of merozoites.** Following Percoll enrichment schizonts were incubated overnight in complete media containing HIS with or without 10 uM E64. After removal of E64, merozoites were allowed to egress. Panel B shows increased proportion of clumped merozoites compared to panel A, which was confirmed by microscopic examination.
(DOCX)

**S11 Fig. Measurement of invasion.** Free merozoites purified by filtration through a 1.2 um filtered were incubated in complete media containing 10% HIS and RBCs at 0.5% hematocrit for a few seconds (A) or 15 minutes (B). The RBCs were washed once and incubated in complete medium overnight. The following day a sample was obtained for flow cytometry and stained

with Hoechst 33342. The apparent percent IRBCs increased from 2.5% to 3.6%. However, microscopic examination showed that the vast majority of these invasive forms were merozoites attached to the surface of RBCs.
(DOCX)

## Acknowledgments

We are grateful to Ronald Taylor, PhD, for the gift of 7C12 anti-C3b monoclonal antibody. We also are grateful for the support of the Penn State College of Medicine Flow Cytometry Core.

## Author Contributions

**Conceptualization:** José A. Stoute, Sergei Biryukov.

**Data curation:** José A. Stoute.

**Formal analysis:** José A. Stoute.

**Funding acquisition:** José A. Stoute.

**Investigation:** José A. Stoute, Mary E. Landmesser, Sergei Biryukov.

**Methodology:** José A. Stoute, Mary E. Landmesser, Sergei Biryukov.

**Project administration:** José A. Stoute, Mary E. Landmesser.

**Resources:** José A. Stoute.

**Software:** José A. Stoute.

**Supervision:** José A. Stoute.

**Validation:** José A. Stoute.

**Visualization:** José A. Stoute.

**Writing – original draft:** José A. Stoute, Sergei Biryukov.

**Writing – review & editing:** José A. Stoute.

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
