## [Decision Letter · Decision Letter 0]

4 Aug 2020

Treatment of Plasmodium falciparum merozoites with the protease inhibitor E64 and mechanical filtration increases their susceptibility to complement activation

PONE-D-20-16732

Dear Dr. Stoute,

We’re pleased to inform you that your manuscript has been judged scientifically suitable for publication and will be formally accepted for publication once it meets all outstanding technical requirements. Although the Reviewer #1 (previous Reviewer #2) has minor suggestions remained, I think you will be able to consider them in your future work.

Kind regards,

Takafumi Tsuboi

Academic Editor

PLOS ONE

Additional Editor Comments (optional):

Reviewers' comments:

Reviewer's Responses to Questions

**Comments to the Author**

1. Is the manuscript technically sound, and do the data support the conclusions?

Reviewer #1: Partly

2. Has the statistical analysis been performed appropriately and rigorously? 

Reviewer #1: Yes

3. Have the authors made all data underlying the findings in their manuscript fully available?

Reviewer #1: Yes

4. Is the manuscript presented in an intelligible fashion and written in standard English?

Reviewer #1: Yes

5. Review Comments to the Author

Reviewer #1: The manuscript has been improved but as discussed at 1st revision the significant differences were very marginal. The baseline samples were prepared without serum (means no contain complement factors), it looks important to normalize 0, 15- and 30-minutes data with baseline data then show the difference statistically. And please add baseline data in Fig 5 -10.

6. PLOS authors have the option to publish the peer review history of their article (what does this mean?). If published, this will include your full peer review and any attached files.

Reviewer #1: No

---

## [Editor Report · Acceptance letter]

11 Aug 2020

PONE-D-20-16732 

 Treatment of Plasmodium falciparum merozoites with the protease inhibitor E64 and mechanical filtration increases their susceptibility to complement activation 

Dear Dr. Stoute:

I'm pleased to inform you that your manuscript has been deemed suitable for publication in PLOS ONE. Congratulations! Your manuscript is now with our production department. 

Kind regards, 

on behalf of

Prof. Takafumi Tsuboi 

Academic Editor

PLOS ONE